# Continuous Adaptive Finite-Time Sliding Mode Control for Fractional-Order Buck Converter Based on Riemann-Liouville Definition

**DOI:** 10.3390/e25040700

**Published:** 2023-04-21

**Authors:** Zhongze Cai, Qingshuang Zeng

**Affiliations:** School of Astronautics, Harbin Institute of Technology, Harbin 150006, China; hitczz@163.com

**Keywords:** fractional calculus, Riemann-Liouville, buck converter, adaptive law, continuous sliding mode control, finite-time stability

## Abstract

This study proposes a continuous adaptive finite-time fractional-order sliding mode control method for fractional-order Buck converters. In order to establish a more accurate model, a fractional-order model based on the Riemann-Liouville (R-L) definition of the Buck converter is developed, which takes into account the non-integer order characteristics of electronic components. The R-L definition is found to be more effective in describing the Buck converter than the Caputo definition. To deal with parameter uncertainties and external disturbances, the proposed approach combines these factors as lumped matched disturbances and mismatched disturbances. Unlike previous literature that assumes a known upper bound of disturbances, adaptive algorithms are developed to estimate and compensate for unknown bounded disturbances in this paper. A continuous finite-time sliding mode controller is then developed using a backstepping method to achieve a chattering-free response and ensure a finite-time convergence. The convergence time for the sliding mode reaching phase and sliding mode phase is estimated, and the fractional-order Lyapunov theory is utilized to prove the finite-time stability of the system. Finally, simulation results demonstrate the robustness and effectiveness of the proposed controller.

## 1. Introduction

The Buck converter is a crucial energy conversion apparatus that assumes a significant role in distributed power supply systems and wind power generation systems [1] by enabling stabilization of the output voltage at the reference output voltage. Consequently, enhancing the performance of the controller has the potential to substantially augment energy conversion efficiency, mitigate energy losses, and improve system stability. However, most current Buck converter models assume that the capacitance and inductance are integer-order, despite the fact that in real systems they are typically non-integer-order. Experimental studies by [2,3] have shown that fractional-order capacitors exist in various dielectrics and have demonstrated that inductors also possess fractional-order characteristics. Using an integer-order model to describe a Buck converter may lead to inaccurate results. Furthermore, the hereditary and memory properties of fractional calculus operators can improve the modeling accuracy and control quality of systems and increase the flexibility of power electronic system design. From the point of modern control theory, the accurate modeling of the controlled object is an important factor in the stability of the control system and can directly affect the performance of the controller. Therefore, researchers have begun to apply fractional calculus to the modeling and control of the Buck converter [4].

Several definitions of fractional calculus, such as R-L, Grunwald–Letnikov, and Caputo, have been proposed in [5,6]. Among them, most studies of the fractional-order model of the Buck converter are based on the Caputo definition. However, due to the differences in definitions, the theoretical results obtained may be significantly different. Moreover, the lower limit of the integral is often set to zero in the Caputo definition to facilitate numerical simulation, which can cause errors. Therefore, some researchers have started to investigate the mathematical model of the Buck converter under the R-L definition. Based on the R-L definition, [7,8] have proposed an equivalent parameter method to analyze and model the Buck converter in both continuous and discontinuous conduction mode. Ref. [9] shows that the overall closed-loop response of the fractional-order Buck converter is more stable as the inductor order decreases. In [10] the R-L fractional-order model of a Buck converter is developed in continuous conduction, which shows more accuracy than the Caputo definition and illustrates the influence of the order of the capacitor/inductor on the modeling of the system. Ref. [11] have concluded that the Buck converter modeled based on the R-L definition exhibits better consistency with practical systems and smaller relative errors in both theoretical and experimental settings, with initial conditions defined with corresponding physical meanings in the circuit.

Traditional control methods have been ineffective in suppressing mismatched disturbances. To address the uncertainties and disturbances in Buck converters and enhance controller performance, researchers have proposed various control strategies, including adaptive control [12], model predictive control [13], robust control [14], and sliding mode control (SMC) [15,16,17]. Among these methods, SMC has garnered significant attention for its inherent robustness and simple structure. However, research on the control of fractional-order Buck converters is currently limited. In [18], adaptive sliding mode control was developed to address matched disturbances and improve the system’s robustness. In [19], a fractional-order terminal sliding mode control was proposed to achieve a finite-time convergence during sliding mode reaching phase. In [20], a fractional-order sliding mode control based on disturbance observer (DOB) was proposed to compensate for mismatched disturbances. Ref. [21] proposes a fractional-order DOB to estimate mismatched disturbance and its derivative and achieve their suppression. Nevertheless, all of the aforementioned studies were based on the Caputo definition. Therefore, exploring controllers designed for R-L definition fractional-order Buck converters could provide novel insights and greater flexibility for circuit system control theory and practice.

Based on the above discussion, this paper proposes a continuous finite-time sliding mode control based on an adaptive law for the fractional-order Buck converter. The main contributions can be concluded as follows:Following the studies in [7,8,9,10,11], a fractional-order Buck converter mathematical model based on R-L definition is developed, which is able to describe the characteristics of the Buck converter more accurately.Compared with the existing works [22,23,24], adaptive laws are developed in this paper to estimate the upper bound of disturbances such that it is not necessary to know the upper bound of the disturbance in advance.Compared with [18,25,26], a globally finite-time stability is achieved in this paper.Compared with [17,20,22,24], a continuous sliding mode control input is developed to attenuate the chattering caused by the traditional discontinuous sign function.

The paper is organized as follows. In Section 2, essential definitions and lemmas of fractional-order calculus are presented. Section 3 derives the fractional-order mathematical model of the Buck converter based on the R-L definition. Section 4 proposes an overall continuous adaptive finite-time sliding mode control strategy using the backstepping method. The effectiveness of the proposed controller is demonstrated through simulation results presented in Section 5. Finally, Section 6 concludes the paper.

## 2. Preliminaries

This section gives the basic concepts of fractional-order calculus and the relevant lemmas.

### 2.1. Fractional Calculus

Fractional calculus redefines the real number order for both integral and derivative calculations. The fractional-order derivatives based on the R-L definition and Caputo definition are introduced in this section.

**Definition 1**.
*The α th-order Caputo fractional derivative for continuous differentiable function f(t) can be defined as*

(1)
$$\begin{array}{c} {}_{t_{0}}^{C}D_{t}^{\alpha }f\left(t\right)=\frac{1}{\Gamma (m-\alpha )}\int_{t_{0}}^{t}\frac{f^{\left(m\right)}\left(\tau \right)}{{\left(t-\tau \right)}^{\alpha -m+1}}d\tau \end{array}$$

*where D denotes the fractional-order calculus operator, α∈[m−1,m), $t>t_{0}$. Γ(·) denotes the Gamma function, which can be represented as*

$$\begin{array}{c} \Gamma \left(\alpha \right)=\int_{0}^{\infty }\tau^{\alpha -1}e^{-\tau }d\tau \end{array}$$

*with α>0.*


**Definition 2**.
*The α th-order R-L fractional derivative for continuous differentiable function f(t) can be given as*

(2)
$$\begin{array}{c} {}_{t_{0}}^{RL}D_{t}^{\alpha }f\left(t\right)=\frac{1}{\Gamma (m-\alpha )}{\left(\frac{d}{dt}\right)}^{m}\int_{t_{0}}^{t}\frac{f(\tau )}{{\left(t-\tau \right)}^{1-m+\alpha }}d\tau \end{array}$$

*where D denotes the fractional-order calculus operator, α∈(m−1,m), $t>t_{0}$.*


It should be noted that a fundamental distinction between the R-L and Caputo definitions resides in the order of differentiation and integration. Specifically, the former proceeds with integration before differentiation on the function f(t), whereas the latter conducts differentiation before integration.

**Remark 1**.
*It is clearly seen that the fractional-order calculus is an extension of the integer-order calculus, which is a special form of the fractional-order calculus. It is therefore necessary to design a controller for the fractional-order model of the Buck converter in order to broaden the range of applications.*


**Remark 2**.
*When f(t) is constant, the fractional-order differentiation outcomes differ under the Caputo and R-L definitions. Specifically, the Caputo differentiation of f(t) yields 0, whereas under the R-L definition, the differential can be expressed as $\frac{C{\left(t-t_{0}\right)}^{-\beta }}{\Gamma (1-\beta )}$. Although scholars typically utilize the Caputo definition with an initial condition set to zero to study power electronic systems, this approach is inaccurate, as noted in [10,11]. In contrast, the R-L definition’s initial condition carries physical significance in the circuit system. The R-L definition of the fractional-order model has been demonstrated to be more precise in describing Buck converters.*


### 2.2. Stability

**Lemma** **1**([27]). *Let V(x)∈R be a continuously differentiable function; then, for $\forall t\geq t_{0}$, the following inequality*
(3)$$\begin{array}{c} \frac{1}{2}{}_{t_{0}}D_{t}^{\alpha }V^{2}\left(x\right)\leq V{\left(x\right)}_{t_{0}}D_{t}^{\alpha }V\left(x\right),\forall \alpha \in \left(0,1\right) \end{array}$$
*holds.*

**Lemma** **2**([28]). *Consider the Caputo or R–L fractional nonautonomous system ${}_{t_{0}}D_{t}^{\alpha }=f\left(t,x\right)$ with $x(t_{0})$, α∈(0,1); $f:\left[t_{0},\infty \right]\times \Omega \to R^{n}$ is piecewise continuous in t and locally Lipschitz in x on $[t_{0},\infty ]\times \Omega$ and $\Omega \in R^{n}$ is a domain that contains the origin x=0. Let x=0 be the equilibrium point for the system. $D\subset R^{n}$ is a domain containing the origin. Suppose V(t,x(t)):[0,∞)×D→R is a continuously differentiable function and locally Lipschitz with respect to x such that*
(4)$$\begin{array}{cc} & \alpha_{1}{∥x∥}^{a}\leq V\left(t,x\left(t\right)\right)\leq \alpha_{2}{∥x∥}^{ab}\\ & D^{\alpha }V\left(t,x\left(t\right)\right)\leq -\alpha_{3}{∥x∥}^{ab} \end{array}$$
*with t≥0, x∈D, α∈(0,1), $\alpha_{1}$, $\alpha_{2}$, $\alpha_{3}$, a and b are arbitrary positive constants. Then x=0 is Mittag–Leffler-stable, and if the assumptions hold globally on $R^{n}$, then the equilibrium point x=0 is globally Mittag–Leffler-stable.*
*Mittag–Leffler-stable implies asymptotically stable.*


**Lemma** **3**([25]). *Suppose a function $g\left(t\right)\in C^{1}\left(\left[0,b\right]\right)$, α∈(0,1), β∈R; then, it obtains*
$$D^{\alpha }g^{\beta }\left(t\right)=\frac{\Gamma (1+\beta )}{\Gamma (1+\beta -\alpha )}g^{\beta -\alpha }\left(t\right)D^{\alpha }g\left(t\right)$$

The symbol *D* in the following sections denotes the R-L fractional-order calculus operator.

## 3. Fractional-Order Mathematical Model of Buck Converter Based on R-L Definition

The Buck converter typically comprises several essential components, such as a voltage source ($V_{in}$), a diode (*D*), an inductance (*L*), a capacitance (*C*), a controller ($S_{\omega }$), and a parasitic resistance (*R*), as depicted in Figure 1.

Without considering disturbances, the mathematical model of the Buck converter with the ON status of $S_{\omega }$ can be written as
(5)$$\begin{array}{c} \left\{\begin{array}{cc} & \frac{di_{L}}{dt}=\frac{1}{L}\left(V_{in}-v_{0}\right)\\ & \frac{dv_{0}}{dt}=\frac{1}{C}\left(i_{L}-\frac{v_{0}}{R}\right) \end{array}\right. \end{array}$$

When it switches to OFF, the model can be written as
(6)$$\begin{array}{c} \left\{\begin{array}{cc} & \frac{di_{L}}{dt}=-\frac{v_{0}}{L}\\ & \frac{dv_{0}}{dt}=\frac{1}{C}\left(i_{L}-\frac{v_{0}}{R}\right) \end{array}\right. \end{array}$$

Combining (Equation 5) and (Equation 6), it obtains
(7)$$\begin{array}{c} \left\{\begin{array}{cc} & \frac{di_{L}}{dt}=\frac{1}{L}\left(\mu V_{in}-v_{0}\right)\\ & \frac{dv_{0}}{dt}=\frac{1}{C}\left(i_{L}-\frac{v_{0}}{R}\right) \end{array}\right. \end{array}$$
where μ denotes the status of $S_{w}$, which takes the value 1 for ON status and 0 for OFF status. The controller determines the value of μ.

Considering the fact that the capacitance and resistance are not of integer-order, to further improve the accuracy of modeling, the fractional-order calculus is introduced here to establish a fractional-order model based on the R-L definition. Rewrite the function (Equation 7) as
(8)$$\begin{array}{c} \left\{\begin{array}{cc} & \frac{d^{\alpha }v_{0}}{dt^{\alpha }}=\frac{1}{C}\left(i_{L}-\frac{v_{0}}{R}\right)\\ & \frac{d^{\beta }i_{L}}{dt^{\beta }}=\frac{1}{L}\left(\mu V_{in}-v_{0}\right) \end{array}\right. \end{array}$$
where α,β∈(0,1) denote the fractional order of capacitance and inductance, respectively, whose values depend on the loss of the capacitance and the proximity effects in the engineering.

Considering the presence of uncertainties and disturbances in the actual system, which may arise from model parameter perturbations and external disturbances, deviations may occur between the actual model and the ideal model. As a result, this paper proposes the development of a mathematical model for the Buck converter, accounting for disturbances and parameter perturbations, expressed as
(9)$$\begin{array}{c} \left\{\begin{array}{cc} & D^{\alpha }v_{0}=\frac{1}{C_{0}+\Delta C}\left(i_{L}-\frac{v_{0}}{R_{0}+\Delta R}\right)+d_{1}\\ & D^{\beta }i_{L}=\frac{1}{L_{0}+\Delta L}\left(\mu \left(V_{in0}+\Delta V_{in}\right)-v_{0}\right)+d_{2} \end{array}\right. \end{array}$$
where $L_{0}$, $C_{0}$, $R_{0}$, $V_{in0}$ are the nominal values of the components of the Buck converter, $\Delta L,\Delta C,\Delta R,\Delta V_{in}$ are the parametric uncertainties of the components, $d_{1}$ and $d_{2}$ are disturbances acting on the current and voltage channels, including unknown dynamics and external disturbances.

**Assumption 1**.
*It is assumed that the disturbances $d_{1}$ are $d_{2}$ are bounded.*


Combining the uncertainties and disturbances in Equation (Equation 9), it obtains
(10)$$\begin{array}{c} \left\{\begin{array}{cc} & D^{\alpha }v_{0}=\frac{1}{C_{0}}\left(i_{L}-\frac{v_{0}}{R_{0}}\right)+d_{1}^{*}\\ & D^{\beta }i_{L}=\frac{1}{L_{0}}\left(\mu V_{in0}-v_{0}\right)+d_{2}^{*} \end{array}\right. \end{array}$$
where $d_{1}^{*},d_{2}^{*}$ are
$$\begin{array}{ccc} d_{1}^{*}\left(t\right) & = & \frac{v_{0}\Delta R}{R_{0}\left(R_{0}+\Delta R\right)\left(C_{0}+\Delta C\right)}+\frac{v_{0}\Delta C-i_{L}\Delta CR_{0}}{C_{0}R_{0}\left(C_{0}+\Delta C\right)}+d_{1}\\ d_{2}^{*}\left(t\right) & = & \frac{\mu \Delta V_{in}L_{0}-\mu \Delta LV_{in0}+\Delta Lv_{0}}{\left(L_{0}+\Delta L\right)L_{0}}+d_{2} \end{array}$$

The objective of this paper is to design a continuous adaptive fractional-order sliding mode controller such that the output of Buck converter $v_{0}$ can track the ideal reference voltage $v_{ref}$ in the presence of matched disturbances and mismatched disturbances.

Let $x_{1}=v_{0}-v_{ref}$; then, the aim is to force $x_{1}\to 0$. Rewrite (Equation 10) as
(11)$$\begin{array}{c} \left\{\begin{array}{cc} & D^{\alpha }x_{1}=x_{2}+w_{1}\\ & D^{\beta }x_{2}=f\left(x_{1},x_{2}\right)+gu+w_{2} \end{array}\right. \end{array}$$
where
$$\begin{array}{ccc} & & x_{1}=v_{0}-v_{ref},x_{2}=\frac{1}{C_{0}}\left(i_{L}-\frac{V_{0}}{R_{0}}\right),g=\frac{V_{in0}}{C_{0}L_{0}},f\left(x_{1},x_{2}\right)=-\frac{1}{C_{0}L_{0}}x_{1}-\frac{1}{L_{0}C_{0}}v_{ref}\\ & & w_{1}=d_{1}^{*}-D^{\alpha }v_{ref},w_{2}=\frac{1}{C_{0}}d_{2}^{*}-\frac{1}{C_{0}R_{0}}D^{\beta }x_{1}-\frac{1}{R_{0}C_{0}}D^{\beta }v_{ref} \end{array}$$

Note that the control gain g>0. There must exist positive constants $K_{1},K_{2}$ such that
$$\begin{array}{c} K_{1}=\underset{t>0}{\sup}\left|w_{1}\right|,K_{2}=\underset{t>0}{\sup}\left|w_{2}\right| \end{array}$$
under the condition of Assumption 1.

**Assumption 2**.
*The disturbances $w_{1}$ and $w_{2}$ are differentiable and their α/β order differentiations are bounded. That is, there exist positive constants $\xi_{1}$,$\xi_{2}$ such that*

$$\begin{array}{c} \xi_{1}=\underset{t>0}{\sup}\left|D^{\alpha }w_{1}\right|,\xi_{2}=\underset{t>0}{\sup}\left|D^{\beta }w_{2}\right| \end{array}$$

*holds.*


## 4. Continuous Adaptive Finite-Time Sliding Mode Control Method

The system described by (Equation 11) is subject to both matched and mismatched disturbances. While the matched disturbance $w_{2}$ directly affects the control channel, linear sliding mode control can effectively suppress its effects and drive the system state to asymptotically converge to the equilibrium point on the sliding surface when $w_{1}=0$. However, when $w_{1}\neq 0$, since it does not directly affect the control channel, the linear sliding mode variable cannot compensate for the effects of the mismatched disturbance as stated in [22]. As a result, the system trajectory may converge to a neighborhood that contains the equilibrium point, with the extent of convergence depending to some extent on the bound of $w_{1}$. Additionally, sudden variations in the disturbances may cause the system state to deviate from the equilibrium point. To address these issues, this paper proposes a novel continuous adaptive sliding mode controller based on the backstepping method to handle unknown bounded disturbances. Adaptive algorithms are developed to estimate the upper bounds of both matched and unmatched disturbances, while a continuous sliding mode controller is designed to suppress chattering.

In accordance with the backstepping method, a virtual control signal $\phi_{2}$ is firstly designed to deal with mismatched disturbances. The system state $x_{2}$ is defined to track the virtual control $\phi_{2}$. $z_{2}$ is the tracking error, which is defined as
(12)$$\begin{array}{c} z_{2}=x_{2}-\phi_{2} \end{array}$$

This easily obtains $x_{2}=z_{2}+\phi_{2}$. By substituting Equation (Equation 12) into (Equation 11), it obtains
(13)$$\begin{array}{c} D^{\alpha }x_{1}=z_{2}+\phi_{2}+w_{1} \end{array}$$

When $z_{2}$ converges to 0, the system state $x_{2}$ can accurately track $\phi_{2}$; rewrite (Equation 13) as
(14)$$\begin{array}{c} D^{\alpha }x_{1}=\phi_{2}+w_{1} \end{array}$$

The new fractional-order sliding mode variable inspired by [29] is proposed as
(15)$$\begin{array}{c} s_{1}=D^{\alpha }x_{1}+C_{1}D^{\alpha -1}\left(x_{1}+x_{1}^{\rho_{1}}\right) \end{array}$$
where $C_{1}$ is a positive constant, $\rho_{1}\in \left(0,1\right)$.

**Theorem 1**.
*Consider the following controller*

(16)
$$\begin{array}{c} \left\{\begin{array}{cc} & \phi_{2}=-C_{1}D^{\alpha -1}\left(x_{1}+x_{1}^{\rho_{1}}\right)+\phi_{n}\\ & D^{\alpha }\phi_{n}=\zeta_{1}-T_{1}\phi_{n}\\ & \zeta_{1}=-\left(\hat{K_{1}}+T_{1}\hat{\xi_{1}}+\eta_{1}{\left|s_{1}\right|}^{\delta_{1}}\right)sign\left(s_{1}\right) \end{array}\right. \end{array}$$

*and adaptive law*

(17)
$$\begin{array}{c} \left\{\begin{array}{ccc} & D^{\alpha }{\hat{K}}_{1}=l_{1}\left|s_{1}\right|,D^{\alpha }{\hat{\xi }}_{1}=T_{1}q_{1}\left|s_{1}\right|, & if\left|s_{1}\right|\geq \Delta_{1}\\ & D^{\alpha }{\hat{K}}_{1}=l_{1}\Delta_{1}sign\left(s_{1}\right),D^{\alpha }{\hat{\xi }}_{1}=T_{1}q_{1}\Delta_{1}sign\left(s_{1}\right), & if\left|s_{1}\right|<\Delta_{1} \end{array}\right. \end{array}$$

*where $\delta_{1}\in \left(0,1\right)$, $\hat{K_{1}}$ and $\hat{\xi_{1}}$ are the estimation of $K_{1}$ and $\xi_{1}$, respectively, $T_{1},q_{1},l_{1},\eta_{1}$ are positive adaptation parameters that play the important role in regulating the adaptation speed. $\Delta_{1}$ is the design constant, which is a very small constant, used to avoid the unbound growth of adaptive gain. When the sliding mode variable is chosen as (Equation 15), then the system (Equation 14) is finite-time stable with the controller (Equation 16) and adaptive law (Equation 17).*


**Proof.** Substitute (Equation 14) and (Equation 16) into (Equation 15); it obtains
(18)$$\begin{array}{ccc} s_{1} & = & \phi_{2}+w_{1}+C_{1}xD^{\alpha -1}\left(x_{1}+x_{1}^{\rho_{1}}\right)\\ & = & \phi_{n}+w_{1} \end{array}$$
Define the Lyapunov function as
(19)$$\begin{array}{c} V_{1}=\frac{1}{2}\left(s_{1}^{2}+q_{1}^{-1}{\tilde{\xi }}_{1}^{2}+l_{1}^{-1}{\tilde{K}}_{1}^{2}\right) \end{array}$$
where ${\tilde{\xi }}_{1}=\xi_{1}-{\hat{\xi }}_{1}$, ${\tilde{K}}_{1}=K_{1}-{\hat{K}}_{1}$. Differentiating (Equation 19) with α-order along (Equation 18) and (Equation 17) based on Lemma 1, one obtains
(20)$$\begin{array}{ccc} D^{\alpha }V_{1} & \leq & s_{1}D^{\alpha }s_{1}+{\tilde{\xi }}_{1}q_{1}^{-1}D^{\alpha }{\tilde{\xi }}_{1}+{\tilde{K}}_{1}l_{1}^{-1}D^{\alpha }{\tilde{K}}_{1}\\ & \leq & s_{1}D^{\alpha }\left(\phi_{n}+w_{1}\right)-{\tilde{\xi }}_{1}q_{1}^{-1}D^{\alpha }{\hat{\xi }}_{1}-{\tilde{K}}_{1}l_{1}^{-1}D^{\alpha }{\hat{K}}_{1}\\ & = & s_{1}\left(\zeta_{1}-T_{1}\phi_{n}+D^{\alpha }\omega_{1}\right)-{\tilde{\xi }}_{1}q_{1}^{-1}D^{\alpha }{\hat{\xi }}_{1}-{\tilde{K}}_{1}l_{1}^{-1}D^{\alpha }{\hat{K}}_{1}\\ & = & \left|s_{1}\right|\left(-{\hat{K}}_{1}-T_{1}{\hat{\xi }}_{1}-\eta_{1}{\left|s_{1}\right|}^{\delta_{1}}\right)+s_{1}\left(-T_{1}\phi_{n}+D^{\alpha }\omega_{1}\right)-{\tilde{\xi }}_{1}q_{1}^{-1}D^{\alpha }{\hat{\xi }}_{1}-{\tilde{K}}_{1}l_{1}^{-1}D^{\alpha }{\hat{K}}_{1}\\ & \leq & \left|s_{1}\right|\left[-\left({\hat{K}}_{1}+T_{1}{\hat{\xi }}_{1}+\eta_{1}{\left|s_{1}\right|}^{\delta_{1}}\right)+T_{1}\left|\phi_{n}\right|+K_{1}\right]-{\tilde{\xi }}_{1}q_{1}^{-1}T_{1}q_{1}\left|s_{1}\right|-{\tilde{K}}_{1}l_{1}^{-1}l_{1}\left|s_{1}\right| \end{array}$$According to (Equation 18) and Assumption A1, we have $s_{1}=\phi_{n}+w_{1}=0\to$$\left|\phi_{n}\right|=\left|\omega_{1}\right|\leq \xi_{1}$. Substituting it into (Equation 20), one has
(21)$$\begin{array}{c} D^{\alpha }V_{1}\leq -\eta_{1}{\left|s_{1}\right|}^{\delta_{1}+1} \end{array}$$
Based on Lemma 2, the state of system (Equation 14) can converge asymptotically to the sliding mode surface $s_{1}=0$. To further study the convergence time of sliding mode reaching phase, define the following auxiliary Lyapunov function:
(22)$$\begin{array}{c} V_{11}=\frac{1}{2}s_{1}^{2}\left(t\right) \end{array}$$
Compared with $V_{1}$ and $V_{11}$, there must exist a positive constant $\eta_{11}>1$ such that
(23)$$\begin{array}{c} \frac{1}{\eta_{11}}{\left(V_{1}\left(t\right)\right)}^{\frac{\delta_{1}+1}{2}}\leq {\left(V_{11}\left(t\right)\right)}^{\frac{\delta_{1}+1}{2}} \end{array}$$
Using (Equation 21), it obtains
(24)$$\begin{array}{ccc} D^{\alpha }V_{1} & \leq & -\eta_{1}{\left|s_{1}\right|}^{\delta_{1}+1}\\ & = & -\eta_{1}{\left(V_{11}^{1/2}\right)}^{\delta_{1}+1}{\left(\sqrt{2}\right)}^{\delta_{1}+1}\\ & = & \left[-\eta_{1}\cdot 2^{\frac{\delta_{1}+1}{2}}\right]\times V_{11}^{\frac{\delta_{1}+1}{2}}\\ & \leq & -{\bar{\eta }}_{1}\cdot {\left(V_{1}\left(t\right)\right)}^{\frac{\delta_{1}+1}{2}} \end{array}$$
where ${\bar{\eta }}_{1}=\frac{2^{\frac{\delta_{1}+1}{2}}\eta_{1}}{\eta_{11}}$. According to Lemma 3, it obtains
(25)$$\begin{array}{ccc} V_{1}\left(t\right)D^{\alpha }V_{1}\left(t\right) & = & \frac{\Gamma (2)}{\Gamma (2+\alpha )}D^{\alpha }\left[V_{1}{\left(t\right)}^{1+\alpha }\right]\\ & \leq & -{\bar{\eta }}_{1}\cdot {\left(V_{1}\left(t\right)\right)}^{\frac{\delta_{1}+3}{2}} \end{array}$$
Let $v\left(t\right)=[V_{1}{\left(t\right)}^{1+\alpha }]$; then, ${\left[V_{1}\left(t\right)\right]}^{\frac{\delta_{1}+3}{2}}={\left[v\left(t\right)\right]}^{\frac{\delta_{1}+3}{2(1+\alpha )}}$. Based on the above calculations, it obtains
(26)$$\begin{array}{c} D^{\alpha }\left[v{\left(t\right)}^{\alpha -\frac{\delta_{1}+3}{2(1+\alpha )}}\right]\leq -{\bar{\eta }}_{1}\frac{\Gamma (1+\alpha -\frac{\delta_{1}+3}{2(1+\alpha )})}{\Gamma (1-\frac{\delta_{1}+3}{2(1+\alpha )})}\frac{\Gamma (2+\alpha )}{\Gamma (2)} \end{array}$$
Taking the fractional integral of both sides of (Equation 26) in (0,t), suppose that $V_{1}\left(t\right)=0$, $\forall t\geq T_{r1}$; then, v(t)=0, and it yields
(27)$$\begin{array}{ccc} & - & v^{\alpha -\frac{\delta_{1}+3}{2(1+\alpha )}}\left(0\right)\\ & \leq & -{\bar{\eta }}_{1}\frac{\Gamma (1+\alpha -\frac{\delta_{1}+3}{2(1+\alpha )})}{\Gamma (1-\frac{\delta_{1}+3}{2(1+\alpha )})}\frac{\Gamma (2+\alpha )}{\Gamma (2)}\frac{t^{\alpha }}{\Gamma (1+\alpha )} \end{array}$$
Then, the value of $T_{r1}$ is obtained as
(28)$$T_{r1}={\left[\frac{V_{1}^{\frac{2\alpha +2\alpha^{2}-\delta_{1}-3}{2}}\left(0\right)\Gamma \left(2\right)\Gamma \left(1-\frac{\delta_{1}+3}{2(1+\alpha )}\right)\Gamma \left(1+\alpha \right)}{{\bar{\eta }}_{1}\Gamma \left(1+\alpha -\frac{\delta_{1}+3}{2(1+\alpha )}\right)\Gamma \left(2+\alpha \right)}\right]}^{\frac{1}{\alpha }}$$
Hence, the state trajectories of the system (Equation 15) will converge to $s_{1}=0$ within a finite time $T_{r1}$.After $s_{1}=0$ is reached, from (Equation 15), it obtains
(29)$$\begin{array}{c} D^{\alpha }x_{1}=-C_{1}D^{\alpha -1}\left(x_{1}+x_{1}^{\rho_{1}}\right) \end{array}$$
Choose the following positive definite function as a Lyapunov function candidate:
(30)$$\begin{array}{c} V_{x_{1}}=\left|x_{1}\right| \end{array}$$
Taking the time derivative of (Equation 30) and using (Equation 29), it obtains
(31)$$\begin{array}{ccc} {\dot{V}}_{x_{1}} & = & sign\left(x_{1}\right){\dot{x}}_{1}\\ & = & sign\left(x_{1}\right)D^{1-\alpha }\left(D^{\alpha }x_{1}\right)\\ & = & sign\left(x_{1}\right)D^{1-\alpha }\left(-C_{1}D^{\alpha -1}\left(x_{1}+x_{1}^{\rho_{1}}\right)\right)\\ & = & -C_{1}\left(\right|x_{1}\left|+\right|x_{1}{|}^{\rho_{1}})\\ & \leq & -{\bar{C}}_{1}\left(\right|x_{1}\left|+\right|x_{1}{|}^{\rho_{1}}) \end{array}$$
with $0<{\bar{C}}_{1}<C_{1}$. After simple calculations, it obtains
(32)$$\begin{array}{ccc} dt & \leq & -\frac{d(|x_{1}|)}{{\bar{C}}_{1}\left(\right|x_{1}\left|+\right|x_{1}{|}^{\rho_{1}})}\\ & = & -\frac{|x_{1}{|}^{-\rho_{1}}d(|x_{1}\left|\right)}{{\bar{C}}_{1}\left(1+\right|x_{1}{|}^{1-\rho_{1}})}\\ & = & -\frac{1}{{\bar{C}}_{1}\left(1-\rho_{1}\right)}\frac{d(|x_{1}{\left|\right)}^{1-\rho_{1}}}{1+|x_{1}{|}^{1-\rho_{1}}} \end{array}$$
Taking the integral of both sides of (Equation 32) from $t_{r}$ to $t_{s}$ and knowing $s_{1}\left(t_{r}\right)=0$ and $x_{1}\left(t_{s}\right)=0$, it obtains
(33)$$\begin{array}{ccc} t_{s}-t_{r} & \leq & -\frac{1}{{\bar{C}}_{1}\left(1-\rho_{1}\right)}\int_{x_{1}\left(t_{r}\right)}^{x_{1}\left(t_{s}\right)}\frac{\left(\right|x_{1}{\left|\right)}^{1-\rho_{1}}}{1+|x_{1}{|}^{1-\rho_{1}}}\\ & = & -\frac{1}{{\bar{C}}_{1}\left(1-\rho_{1}\right)}ln(1+|x_{1}{|}^{1-\rho_{1}}){|}_{x_{1}\left(t_{r}\right)}^{x_{1}\left(t_{s}\right)}\\ & = & \frac{1}{{\bar{C}}_{1}\left(1-\rho_{1}\right)}ln(1+|x_{1}\left(t_{r}\right){|}^{1-\rho_{1}}) \end{array}$$
where $t_{s}$ denotes the convergence time from $x_{0}$ to x=0 and $t_{r}$ denotes the convergence time from $s(x_{0})$ to s=0. Therefore, the state $x_{1}$ will converge to zero along the sliding mode surface in the finite time $t=t_{s}-t_{r}\leq \frac{1}{{\bar{C}}_{1}\left(1-\rho_{1}\right)}ln(1+|x_{1}\left(t_{r}\right){|}^{1-\rho_{1}})$. Thus, the overall finite-time stability of the system (Equation 15) under controller (Equation 16) is proved. □

Secondly, the control *u* is designed to force the system state $x_{2}$ to track the virtual control $\phi_{2}$, that is, $z_{2}\to 0$. Taking the β-order time-derivative on both sides of the Equation (Equation 12), it obtains
(34)$$\begin{array}{ccc} D^{\beta }z_{2} & = & D^{\beta }x_{2}-D^{\beta }\phi_{2}\\ & = & f+gu+w_{2}-D^{\beta }\phi_{2} \end{array}$$
For system (Equation 34), a new sliding mode variable is designed as
(35)$$\begin{array}{c} s_{2}=D^{\beta }z_{2}+C_{2}D^{\beta -1}\left(z_{2}+z_{2}^{\rho_{2}}\right) \end{array}$$
where $C_{2}$ denotes a positive constant and $\rho_{2}\in \left(0,1\right)$.

**Theorem 2**.
*Consider the following controller*

(36)
$$\begin{array}{c} \left\{\begin{array}{cc} & u=g^{-1}\left(-f+D^{\beta }\phi_{2}-C_{2}D^{\beta -1}\left(z_{2}+z_{2}^{\rho_{2}}\right)+u_{n}\right)\\ & D^{\beta }u_{n}+T_{2}u_{n}=\zeta_{2}\\ & \zeta_{2}=-\left(\hat{K_{2}}+T_{2}\hat{\xi_{2}}+\eta_{2}{\left|s_{2}\right|}^{\delta_{2}}\right)sign\left(s_{2}\right) \end{array}\right. \end{array}$$

*and adaptive law*

(37)
$$\begin{array}{c} \left\{\begin{array}{ccc} & D^{\beta }{\hat{K}}_{2}=l_{2}\left|s_{2}\right|,D^{\beta }{\hat{\xi }}_{2}=T_{2}q_{2}\left|s_{2}\right|, & if\left|s_{2}\right|\geq \Delta_{2}\\ & D^{\beta }{\hat{K}}_{2}=l_{2}\Delta_{2}sign\left(s_{2}\right),D^{\beta }{\hat{\xi }}_{2}=T_{2}q_{2}\Delta_{2}sign\left(s_{2}\right), & if\left|s_{2}\right|<\Delta_{2} \end{array}\right. \end{array}$$

*where $l_{2}$, $q_{2}$, $T_{2}$, $\eta_{2}$ are positive constants, $\delta_{2}\in \left(0,1\right)$, and $\Delta_{2}$ is the design constant, which is a very small constant, used to avoid the unbound growth of adaptive gain. When the sliding mode variable is chosen as (Equation 35), then the system (Equation 34) is finite-time stable with the controller (Equation 36) and adaptive law (Equation 37).*


**Proof.** Substituting (Equation 34) and (Equation 36) into (Equation 35), it obtains
(38)$$\begin{array}{ccc} s_{2} & = & u_{n}+w_{2} \end{array}$$
Define the following Lyapunov function as
(39)$$\begin{array}{c} V_{2}=\frac{1}{2}\left(s_{2}^{2}+q_{2}^{-1}{\tilde{\xi }}_{2}^{2}+l_{2}^{-1}{\tilde{K}}_{2}^{2}\right) \end{array}$$
where ${\tilde{\xi }}_{2}=\xi_{2}-{\hat{\xi }}_{2}$, ${\tilde{K}}_{2}=K_{2}-{\hat{K}}_{2}$. Differentiating (Equation 39) with β-order along (Equation 36) and (Equation 35) based on Lemma 1, it obtains
(40)$$\begin{array}{ccc} D^{\beta }V_{2} & \leq & s_{2}D^{\beta }s_{2}+{\tilde{\xi }}_{2}q_{2}^{-1}D^{\beta }{\tilde{\xi }}_{2}+{\tilde{K}}_{2}l_{2}^{-1}D^{\beta }{\tilde{K}}_{2}\\ & \leq & s_{2}D^{\beta }\left(u_{n}+\omega_{2}\right)-{\tilde{\xi }}_{2}q_{2}^{-1}D^{\beta }{\hat{\xi }}_{}-{\tilde{K}}_{2}l_{2}^{-1}D^{\beta }{\hat{K}}_{2}\\ & = & s_{2}\left(\zeta_{2}-T_{2}u_{n}+D^{\beta }\omega_{2}\right)-{\tilde{\xi }}_{2}q_{2}^{-1}D^{\beta }{\hat{\xi }}_{2}-{\tilde{K}}_{2}l_{2}^{-1}D^{\beta }{\hat{K}}_{2}\\ & = & \left|s_{2}\right|\left(-{\hat{K}}_{2}-T_{2}{\hat{\xi }}_{2}-\eta_{2}{\left|s_{2}\right|}^{\delta_{2}}\right)+s_{2}\left(-T_{2}u_{n}+D^{\beta }\omega_{2}\right)-{\tilde{\xi }}_{2}q_{2}^{-1}D^{\beta }{\hat{\xi }}_{2}-{\tilde{K}}_{2}l_{2}^{-1}D^{\beta }{\hat{K}}_{2}\\ & \leq & \left|s_{2}\right|\left[-\left({\hat{K}}_{2}+T_{2}{\hat{\xi }}_{2}+\eta_{2}{\left|s_{2}\right|}^{\delta_{2}}\right)+T_{2}\left|u_{n}\right|+K_{2}\right]-{\tilde{\xi }}_{2}q_{2}^{-1}T_{2}q_{2}\left|s_{2}\right|-{\tilde{K}}_{2}l_{2}^{-1}l_{2}\left|s_{2}\right| \end{array}$$When the system states move on the sliding mode surface according to (Equation 38), it obtains $s_{2}=0$, then $\left|u_{n}\right|=\left|d_{2}\right|\leq \xi_{2}$. Substituting it into the above function, it yields
(41)$$\begin{array}{c} D^{\beta }V_{2}\leq -\eta_{2}{\left|s_{2}\right|}^{\delta_{2}+1}. \end{array}$$
Similar to Theorem 1, the asymptotic stability of system (Equation 35) is guaranteed based on Lemma 2. The deduction of convergence time is the same as Theorem 1 and is thus omitted here. The estimation of convergence time of sliding mode reaching phase $T_{r2}$ for system (Equation 35) is obtained as
(42)$$T_{r2}={\left[\frac{V_{2}^{\frac{2\beta +2\beta^{2}-\delta_{2}-3}{2}}\left(0\right)\Gamma \left(2\right)\Gamma \left(1-\frac{\delta_{2}+3}{2(1+\beta )}\right)\Gamma \left(1+\beta \right)}{{\bar{\eta }}_{2}\Gamma \left(1+\beta -\frac{\delta_{2}+3}{2(1+\beta )}\right)\Gamma \left(2+\beta \right)}\right]}^{\frac{1}{\beta }}$$
The estimation of convergence time on the sliding mode phase is
(43)$$T_{s2}\leq \frac{1}{{\bar{C}}_{2}\left(1-\rho_{2}\right)}ln(1+|z_{2}\left(t_{r}\right){|}^{1-\rho_{2}})+T_{r2}$$
with ${\bar{C}}_{2}\in \left(0,C_{2}\right)$. This completes the proof. □

On the basis of Theorems 1 and 2, the finite-time stability of the overall system (Equation 11) is guaranteed. The overall block diagram of the Buck converter control system is shown in Figure 2.

**Remark 3**.
*Figure 2 demonstrates the attainment of global finite-time stability for the system. Initially, the designed controller *u* ensures that $s_{2}=0$. Subsequently, during the sliding mode phase, the error signal $z_{2}$ is forced to 0, resulting in precise tracking of the virtual control signal $\phi_{2}$ by the system state $x_{2}$. Once the sliding mode variable $s_{1}$ reaches 0 within finite time, the system output $y=x_{1}$ is stabilized at 0 under the virtual controller $\phi_{2}$.
*


## 5. Simulation

In order to validate the effectiveness and applicability of the proposed continuous adaptive finite-time sliding mode controller, this section employs the Matlab/Simulink simulation platform and the FOTF toolbox to establish the mathematical model of the fractional-order Buck converter based on the R-L definition. The results are analyzed. The parameters of the Buck converter and reference output voltage are shown in Table 1.

Considering uncertainties and disturbances that exist in the Buck converter and without loss of generality, the matched and mismatched disturbances are set as $\omega_{1}=2.5\sin\left(t\right)+0.5+1.2\cos\left(t\right)$ and $\omega_{2}=1.4\cos\left(t\right)$ to verify the robustness of the proposed controller. The control object is to track the reference voltage of the Buck converter $v_{ref}$ against disturbances. Table 2 shows the parameters of the controller. According to the above discussion, the parameters α=0.9 and β=0.95 are selected to obtain a more accurate simulation result.

The initial state values of system (Equation 10) are set to [−15,0] in accordance with the definition of state variable $x_{1}$. The simulation results for the system output voltage $v_{0}$, state variables $x_{1}$ and $x_{2}$, and the tracking state $z_{2}$ are presented in Figure 3. The results indicate that the proposed controller is capable of accurately and rapidly tracking the reference output voltage under both matched and mismatched disturbances, and can maintain system stability under nonvanishing disturbances, thereby showcasing its high performance and robustness. However, it should be noted that the system state $x_{2}$ is not stabilized at 0 due to the presence of mismatched disturbances. To address this issue, the proposed sliding mode controller adopts the backstepping method and introduces a virtual control variable $\phi_{2}$, which is forced to track $x_{2}$. In doing so, $x_{2}$ can be employed to suppress the mismatched disturbance $w_{1}$ in the system and force $z_{2}\to 0$. Despite the nonvanishing disturbances set in the simulation, $x_{2}$ can track $-w_{1}$ under the controller *u*, thus enabling $\dot{x}1\to 0$ and achieving the tracking of system output $v_{0}$ to vref. In addition, Figure 4 illustrates the two sliding mode variables $s_{1}$ and $s_{2}$ that are designed in the controller. It can be observed that the two control laws designed can make the state points reach the sliding surfaces in finite time, thus verifying the finite-time stability of the Buck converter and the robustness of the proposed controller.

Figure 5 shows the partial control signal of the sliding mode controller. It is evident that the actual control signal *u* is smooth. Taking the controller (Equation 16) as an example, the chattering phenomenon of sliding mode control stems from the discontinuity of the control, that is, the sign function. The discontinuous control signal causes the discontinuous chattering output. This paper proposes a controller inspired by the idea of the super-twisting algorithm, placing the discontinuous term in the $\zeta_{2}$ term and integrating it to obtain a continuous actual control signal. The $\zeta_{2}$ signal is discontinuous, but the $u_{n}$ signal after being filtered by a fractional-order integral filter is smoothed, which can reduce the chattering while enhancing the robustness of the system and maintaining the effectiveness of sliding mode controller. This illustrates the continuous property of the proposed controller.

Figure 6 shows the disturbance observation values obtained from the adaptive algorithms. It can be seen that the parameters $\hat{K_{1}}$, $\hat{K_{2}}$, $\hat{\xi_{1}}$, and $\hat{\xi_{2}}$ obtained by the adaptive algorithms (Equation 17) and (Equation 37) can all converge to certain constants within a finite time. In the simulation, the mismatched disturbance term is $\omega_{1}=2.5sin\left(t\right)+0.5+1.2cos\left(t\right)$. The estimated value of $\hat{K_{1}}$ obtained by the adaptive algorithm approaches around 4, while the value of $\hat{\xi_{1}}$ obtained by the algorithm is significantly reduced due to the constant disturbance term, as shown in the figure, approaching around 2.2. The matched disturbance term is $\omega_{2}=1.4cos\left(2t\right)$, and the estimated value of $\hat{K_{2}}$ obtained by the adaptive algorithm approaches around 2.1, while the obtained $\hat{\xi_{2}}$ approaches around 3.1. The above discussion illustrates that the proposed adaptive algorithm in this paper is effective and can estimate the upper bound of disturbances in the presence of unknown bounded disturbances, allowing the system output to track the reference voltage under the proposed controller and adaptive law.

To test the robustness against different kinds of disturbances, sudden changed time-varying disturbances and random disturbance are included and the result can be seen in Figure 7. Plots of (a) are under the following mismatched disturbance: (44)$$\begin{array}{c} w_{1}=\left\{\begin{array}{cc} & 2.5sin(t)+0.5+1.2cos(t),\;\;t<2\;and\;t\geq 5\\ & 1.5sin(t)+1.5+0.5cos(t),\;\;t\in [2,5) \end{array}\right. \end{array}$$ The matched disturbance $w_{2}$ keeps the same as above. It is clearly seen that the system state $x_{1}$ is stabilized at 0 under the proposed controller, and $x_{2}$ follows the changed $-w_{1}$ rapidly. Plots of (a) are under $\omega_{1}=$2.5sin(t)+0.5 + 1.2cos(t) and $\omega_{2}=$1.4cos(t) and when t∈(2,5), random disturbances conform to a normal distribution with standard deviation and mean square error set as (0,1). $x_{0}$ is chattering around 0 but acceptable due to the random varying disturbances as in (b). Figure 7 validates the robustness of the proposed controller against multiple disturbances.

In order to further validate the effectiveness of the proposed controller, a comparative analysis was conducted with various existing controllers, including traditional sliding mode control (TSMC), fractional-order disturbance-based complementary sliding mode control (FDOB-CSMC) proposed in [30], fractional-order disturbance-based SMC (FDOB-SMC) presented in [21], and asymptotically stable adaptive continuous SMC (AS-ACSMC) proposed in [25]. The comparison was conducted under identical conditions, and the results are presented in Figure 8.

It is evident from Figure 8 that all the considered control methods achieve convergence, but traditional sliding mode control (TSMC) is unable to effectively suppress mismatched disturbances, resulting in a significantly higher steady-state error than the other methods. Additionally, the convergence speed of TSMC is highly dependent on the sliding mode surface coefficient $k_{t}$, as noted in [20]. The larger the coefficient value, the faster the variation, which increases the system’s chattering and requirements for the controller, potentially leading to degradation of control quality in practical systems. In contrast, fractional-order disturbance-based complementary sliding mode control (FDOB-CSMC) introduces DOB to suppress mismatched disturbances, but its steady-state error is still higher than that of the proposed adaptive sliding mode algorithm. Moreover, FDOB-CSMC requires prior knowledge of the disturbance upper bound, which is challenging to obtain in practical systems and can cause significant overshoots, conflicting with the emphasis on stability in the Buck converter system. Similarly, the FDOB-SMC utilizes a fractional-order DOB with easy structure and observer-based sliding mode variable design, but its convergence speed is slow, and it exhibits high overshoot due to the simpler structure of DOB. The parameters of FDOB-SMC as presented in Theorem 1 of [21] are $l_{1}=10,l_{2}=8,c_{1}=10,\xi_{1}=5$, and $\xi_{2}=3$. Both DOB-based methods show unacceptable overshoot since the initial value of $x_{1}$ in the Buck converter system is –15, which necessitates a strong adjustment speed of DOB. Therefore, the proposed adaptive continuous sliding mode control (ACSMC) algorithm balances the large overshoot and steady-state error brought by the controller and can effectively utilize the adaptive algorithm to estimate the disturbance upper bound, achieving good control performance. Compared with asymptotically stable adaptive continuous SMC (AS-ACSMC) in [25], the proposed finite-time controller exhibits a faster convergence speed, which is crucial for applications.

It is worth noting that both CSMC and TSMC methods employ a disturbance observer to handle mismatched disturbances; however, these methods require prior knowledge of the upper bound of the disturbances. Unfortunately, in many applications, obtaining such knowledge is not feasible. In contrast, the proposed adaptive law in this paper enables estimation of the upper bound of disturbances and can effectively suppress disturbances that are unknown but bounded, thereby allowing more flexibility in controller design. The superiority of this adaptive controller is further elucidated.

## 6. Conclusions

This article proposes a novel mathematical model based on the R-L fractional calculus definition for the Buck converter that takes into account the challenges presented by practical systems, including the existence of non-integer-order components and uncertainties/disturbances in practical systems. The lumped disturbances of the system are separated into matched and mismatched disturbances. Considering the different differential orders of capacitance and inductance and the unknown upper bound disturbances, adaptive laws are developed to estimate the disturbance upper bound and suppress them. The proposed continuous adaptive sliding mode controller based on the backstepping method is an effective solution for the Buck converter system with both matched and mismatched disturbances. By introducing a virtual control variable and designing an adaptive algorithm, the controller can compensate for the unknown bounded disturbances and ensure the system’s robustness and stability. The global finite-time stability property of the proposed controller improves the convergence speed and guarantees the system’s stability within a finite time. Moreover, the proposed controller’s output signal is continuous, which significantly reduces the chattering phenomenon commonly seen in sliding mode control systems. The simulation results demonstrate that the proposed controller can ensure the Buck converter output to track the reference voltage rapidly and precisely, even under the influence of nonvanishing disturbances. Additionally, the adaptive algorithm shows effectiveness in estimating and handling disturbances. Comparison shows the superiority of the proposed controller.

## Figures and Tables

**Figure 1 entropy-25-00700-f001:**
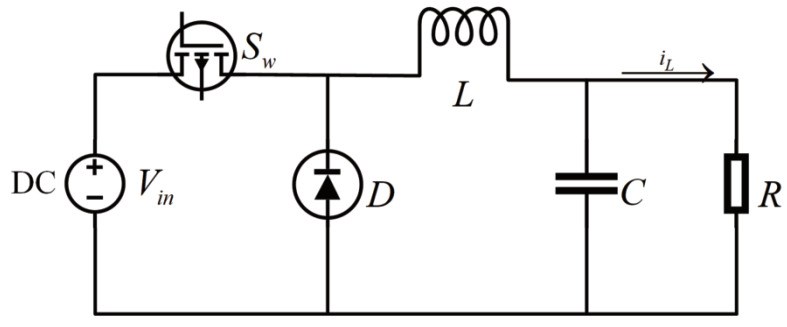
Block diagram of Buck converter.

**Figure 2 entropy-25-00700-f002:**
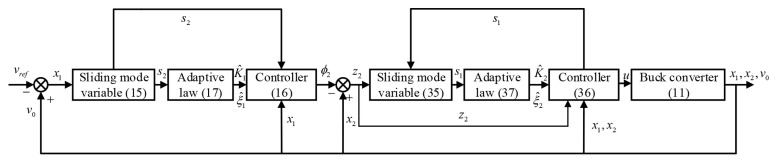
Block diagram of Buck converter control system.

**Figure 3 entropy-25-00700-f003:**
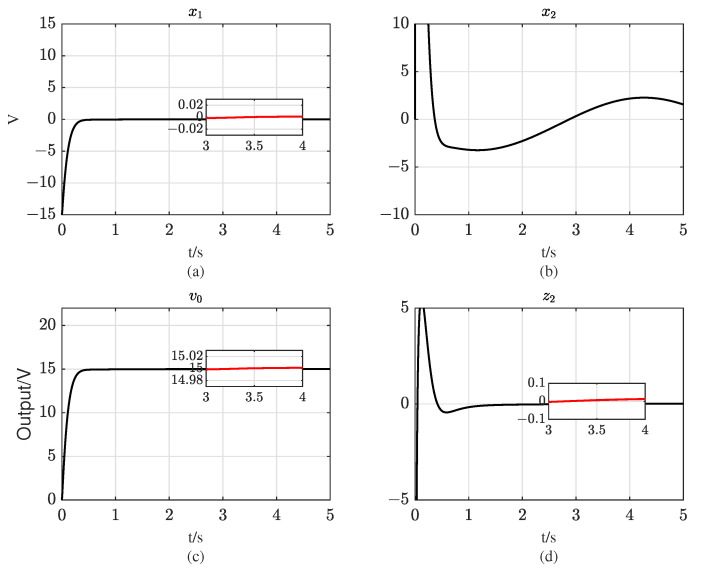
Tracking curves of Buck converter under the proposed controller: (**a**) curve of system state $x_{1}$; (**b**) curve of system state $x_{2}$; (**c**) curve of system output $v_{0}$; (**d**) curve of virtual state $z_{2}$.

**Figure 4 entropy-25-00700-f004:**
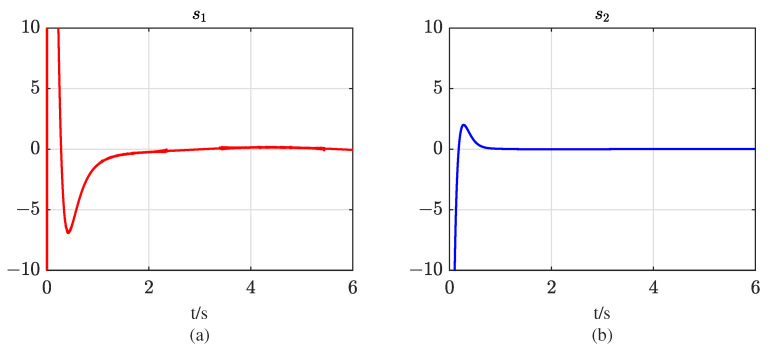
Curves of sliding mode variables: (**a**) curve of sliding mode variable $s_{1}$; (**b**) curve of sliding mode variable $s_{2}$.

**Figure 5 entropy-25-00700-f005:**
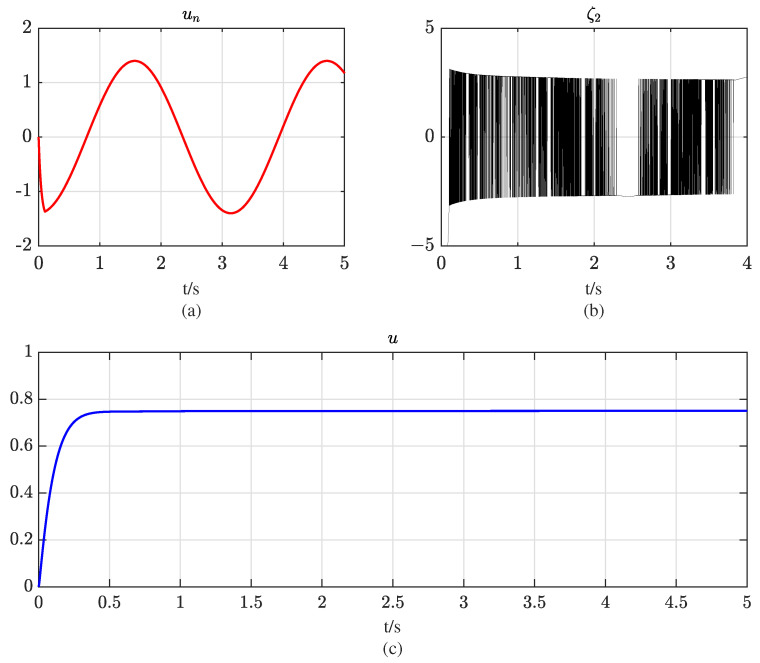
Curves of the actual control signal and virtual signals: (**a**) curve of $u_{n}$; (**b**) curve of $\xi_{2}$; (**c**) curve of the actual control input *u*.

**Figure 6 entropy-25-00700-f006:**
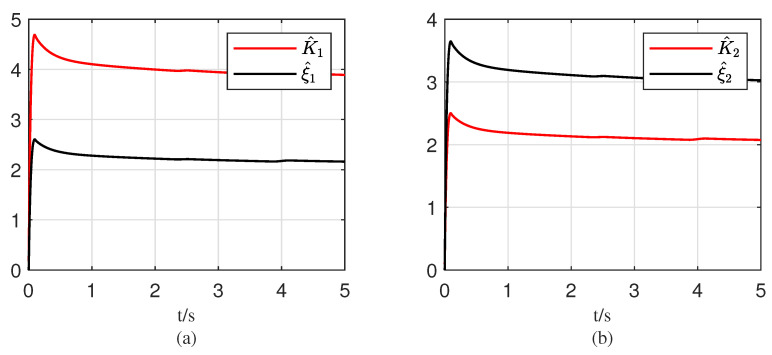
Curves of the estimated values of disturbances $w_{1}$ and $w_{2}$ based on the proposed adaptive law: (**a**) estimations of adaptive law (Equation 17); (**b**) estimations of adaptive law (Equation 37).

**Figure 7 entropy-25-00700-f007:**
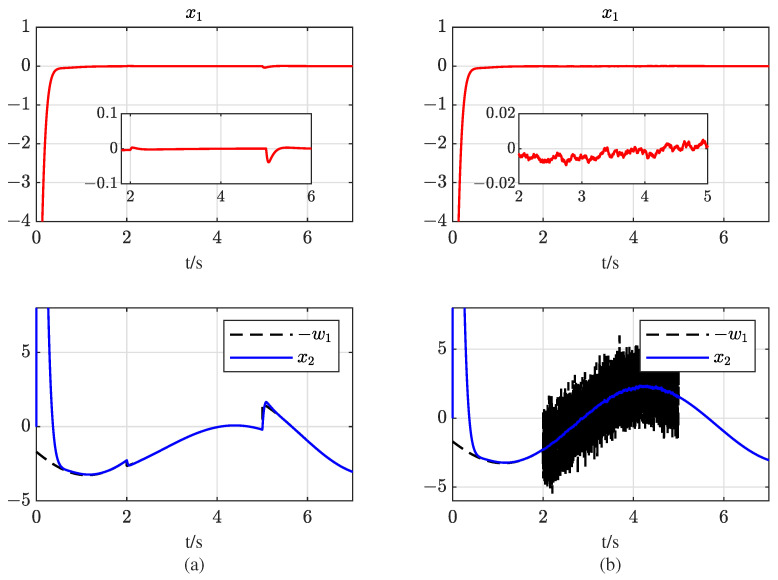
Plots of system states under different disturbances: (**a**) with sudden changed disturbance; (**b**) with random disturbance.

**Figure 8 entropy-25-00700-f008:**
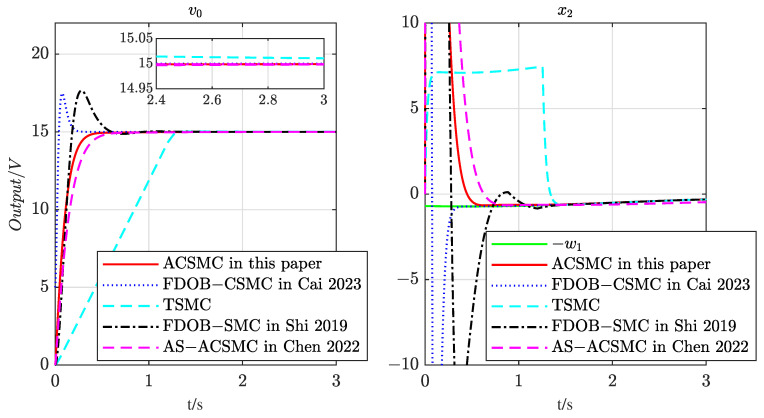
Output voltage of the Buck converter using the controller proposed in this paper compared with the method stated in [21,25,30].

**Table 1 entropy-25-00700-t001:** Parameters of Buck converter.

Description	Parameter	Units	Nominal Value
Load resistance	$R_{0}$	Ω	100
Inductor	$L_{0}$	mH	2.0
Capacitor	$C_{0}$	mF	1.1
Input voltage	$V_{in}$	V	20
Reference voltage	$V_{ref}$	V	15

**Table 2 entropy-25-00700-t002:** Parameters of continuous adaptive finite-time sliding mode controller.

Description	Parameter	Description	Parameter
$C_{1}$	10	$C_{2}$	10
$\rho_{1}$	0.5	$\rho_{2}$	0.5
$T_{1}$	0.1	$T_{2}$	0.1
$q_{1}$	100	$q_{2}$	80
$l_{1}$	40	$l_{2}$	100
$\eta_{1}$	18	$\eta_{2}$	20
$\delta_{1}$	0.8	$\delta_{2}$	0.9

## Data Availability

The data that support the findings of this study are available on request from the corresponding author upon reasonable request.

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
