# Peer review of "Continuous Adaptive Finite-Time Sliding Mode Control for Fractional-Order Buck Converter Based on Riemann-Liouville Definition"

_entropy, 2023, doi:10.3390/e25040700_

Round 1

Reviewer 1 Report

This paper investigates a continuous adaptive finite-time fractional-order sliding mode control method for fractional-order Buck converter. To handle parameter uncertainties and external disturbances, the proposed approach combines these factors as lumped matched disturbances and mismatched disturbances, and unlike some literature that assumes a known upper bound of disturbances, adaptive algorithms are developed to estimate and compensate for unknown bounded disturbances in this paper. A continuous finite-time sliding mode controller is then developed to achieve a chattering-free response and ensure a finite-time convergence utilizing a backstepping method. The idea is interesting. The following comments are given: 1. How to ensure K_{1} and K_{2} in (15) in practice should be discussed. 2. The rationality of Assumption 2 should be discussed. 3. From (41),  how to obtain the setting time in (42)? 4. How to avoid the chattering problem should be further discussed. 5. It is better to give the experiment to verify the effectiveness of proposed algorithm.

Author Response

  1. How to ensure K_{1} and K_{2} in (15) in practice should be discussed.

Thank you very much for your comments here. As shown in chapter 3, the authors state a detailed description of the derivation process. As (9), disturbance includes d1, d2, and some parameters uncertainties. This paper lumped all the disturbances into the lumped matched/mismatched disturbances as (11) and suppose they have an unknown but bounded upper. The authors refer to several article, for emample,

YIN Y. Advanced control strategies for DC– DC buck converters with parametric uncertainties via experimental evaluation, IEEE Transactions on Circuits System, 2020, 67(12): 5257 – 5267. See the sentence after (32) and assumption 1.

LIN X, LIU Y, LIU F, Z. Liu, Y. Fractional-Order sliding mode approach of Buck converters with mismatched disturbances. IEEE Transactions on Circuits and Systems, 2021, 68(9): 3890 – 3900. See assumption 1.

Ding S, Zheng W X, Sun J, et al. Second-order sliding-mode controller design and its implementation for buck converters[J]. IEEE Transactions on Industrial Informatics, 2017, 14(5): 1990-2000. See the sentence between (3) and (4) on page 1991.

To the best knowledge of the authors, almost all researchers directly suppose a bounded upper of the disturbances, thus proving its stability. Some researchers utilize observers to estimate the disturbances and it requires a known upper-bounded disturbance. For the buck converter system, as deduced in Chapter 3, the parameter uncertainties/ external disturbances should and will be bounded because stability is the most important thing from the control viewpoint. This demonstrates the rationality of Assumptions 1 and 2 in this paper. Please the reviewer understands this point. Thank you!

  1. The rationality of Assumption 2 should be discussed.

Please see answer 1.

  1. From (41), how to obtain the setting time in (42)?

Thank you very much for your comments here. Actually, the authors do omit the key description of the obtain of the settling time here. The derivation process is completely the same as Theorem 1 from (22) to (28). As the sentence in line 244-246 shows, ‘The deduction of convergence time is the same as Theorem 1 (from line 206 to line 216), thus omitted here.’ To save some space.

  1. How to avoid the chattering problem should be further discussed.

Chattering is a very common and basic problem exists in sliding mode control, arising from the discontinuity of the control signal, for example, the sign function, which can cause the system output to chatter up and down along the sliding mode plane. The proposed sliding mode controller is continuous, as partly shown in (36), the actual control signal u is obvious continuous because the authors borrow from the idea of the super twisting algorithm, the sign function is placed in xi2, thus generating a continuous control input (also shown in Fig5, xi2 is chattering and un is continuous). Therefore it reduces the chattering caused by the discontinuous control input. This is one of the important innovations of this paper as the summary in line 78. The authors add some essential words here to further introduce why it is chattering. Please refer to it.

  1. It is better to give the experiment to verify the effectiveness of proposed algorithm.

Thank you very much for your comments and professional advice here.

The innovative points of this paper, include the establishment of a more accurate mathematical model of Buck based on the R-L definition and a new controller for this more accurate mathematical model, improvements in both the sliding mode variables and the adaptive algorithm, continuous control signals to solve the chattering problem, simulations to validate the controller designed in this paper and proof of global finite time convergence. The authors appreciate your valuable comments, however, the innovation of this paper, as mentioned above, aims to improve the control performance in two aspects, modeling accuracy, and controller performance, and solve the problem of mismatch interference and verify the arguments of this paper from simulations.
Regarding the experiments you mentioned, the authors have recently been discussing with our team to build an experimental platform in collaboration with relevant teams to further this research and add some experiments to further verify the controller. Unfortunately, at present, there are indeed no conditions for the authors to complete a physical experiment due to objective constraints. From the simulation point of view, the numerical simulation of this paper demonstrates the robustness and better performance of the overall control scheme proposed in this paper which can demonstrate the effectiveness of the proposed method theoretically and from a simulation view. It is sufficient to support the point of this article.

The authors follow the idea of rigorous theoretical analysis and simulation verification before physical experiments. The authors will do some following research on physical experiments in the following work.

Reviewer 2 Report

Authors investigate a continuous adaptive finite-time fractional-order sliding mode

control method for fractional-order Buck converter.

L.160: in the formula of Assumption 2, there is a comma missing.

L.189: choosen should be chosen. Please, revise English.

L.303: a reference is missing.

Questions:

- What happens in case of random disturbances in Assumption 1? Please, comment about this topic. For example,

see Shi, Xia, Liu, Rees On Designing of Sliding-Mode Control for Stochastic Jump Systems, IEEE Transactions on Automatic Control 51(1):97 - 103, 2006.

- What happens in a co-dimension 2 case? More generally, what happens if one uses Filippov or Utkin

approaches, which differe in case vector fields are nonlinear? It is known that Filippov and Utkin systems are important for

these applications, see AF Filippov, Differential Equations with Discontinuous Righthand Sides

Control Systems, Springer 1988 and VI Utkin, Sliding Modes in Control and Optimization, Springer 1992.

Do techniques in FV Difonzo, A note on attractivity for the intersection of two discontinuity manifolds, Opuscula Mathematica 40, no. 6 (2020), 685-702

apply? A comment or an example about this would be valuable, since it seems that nonlinearity is overlooked in the paper.

The paper is interesting, but needs to consider more applications and literature on the topic to be robust enough.

Author Response

Comments and Suggestions for Authors

L.160: in the formula of Assumption 2, there is a comma missing.

Thank you very much for your comments here. The authors correct the mistake here. Apologize for the carelessness.

L.189: choosen should be chosen. Please, revise English.

Thank you very much for your comments here. The authors correct the mistake here. Apologize for the carelessness.

L.303: a reference is missing.

Thank you very much for your comments here. The authors correct the mistake here. Apologize for the carelessness.

1

What happens in case of random disturbances in Assumption 1? Please, comment about this topic. For example, see Shi, Xia, Liu, Rees On Designing of Sliding-Mode Control for Stochastic Jump Systems, IEEE Transactions on Automatic Control 51(1):97 - 103, 2006.

Thank you very much for your professional advice here. The authors refers to the above literature you mentioned. The literature mainly studies the sliding mode controller for Markovian jump systems with some stochastic stability. To be honest, it is beyond the author’s research area. We search the key words ‘Buck converter’ and ‘Markovian’. Only 2 articles are related to the two key words.

Almost Sure Finite-Time Control for Markovian Jump Systems Under Asynchronous Switching With Applications: A Sliding Mode Approach, 2022, Wang Y, Xu SY, IEEE TRANSACTIONS ON CIRCUITS AND SYSTEMS. It seems that this paper uses a Buck converter as a simulation plant and donot discuss the connection between Buck converter and Markovian Jump Systems and it concerns more about the time-varying delays and asynchronous switching which is completely different from this paper. Note that the plant is integer-order.

  1. Vargas, L. P. Sampaio, L. Acho, L. Zhang and J. B. R. do Val, "Optimal Control of DC-DC Buck Converter via Linear Systems With Inaccessible Markovian Jumping Modes," in IEEE Transactions on Control Systems Technology, vol. 24, no. 5, pp. 1820-1827, Sept. 2016, doi: 10.1109/TCST.2015.2508959. This paper mainly concerna about the average cost control problem and added uncertainties to source voltage and load driven by a homogeneous Markov chain. This paper seems interesting. It may provide an idea combined with Markov jump disturbances and Buck converter. Note that the plant is integer-order. Compare with my paper, the main object is different and assumptions of bounded disturbances is different. The authors appreciate your valuable advice and will follow this idea to further the future works. The author think the control method proposed in this paper is supported by strict stability and the research ground is suitable. Your review is very interesting thus the authors adds additional simulations into the manuscript, Figure 7 and 8, to further prove the robustness of the the controller include sudden change disturbances some stochastic generated but bounded disturbances and new comparision controller to validate the superior of the proposed method. The author accept your kind advice and add additional simulations to further validate the effectiveness of the proposed controller. Please check newly added Figure 7 and 8. The authors design new types of disturbances to test the robustness of the controller. Also, the other methods proposed by some other literature are discussed and the simulation result can be seen clearly that the controller proposed in this paper show superior performance and robustness compared with the others. Please check.

2

What happens in a co-dimension 2 case? More generally, what happens if one uses Filippov or Utkin approaches, which differe in case vector fields are nonlinear? It is known that Filippov and Utkin systems are important for these applications, see AF Filippov, Differential Equations with Discontinuous Righthand Sides Control Systems, Springer 1988 and VI Utkin, Sliding Modes in Control and Optimization, Springer 1992.

Please see problem 3.

3

Do techniques in FV Difonzo, A note on attractivity for the intersection of two discontinuity manifolds, Opuscula Mathematica 40, no. 6 (2020), 685-702 apply? A comment or an example about this would be valuable, since it seems that nonlinearity is overlooked in the paper.

Thank you very much for your comments and professional advice here. The authors do refer to the above literature you mentioned. As we know, Utkin is the first researcher who proposed the concept of sliding mode control and the basic traditional sliding mode method is understood in Filippov sense. As the literature you mentioned, the two monographs, How to solve the differential equations with discontinuous right-hand sides and Dynamics is the foundation of sliding mode control. This paper proposed a continuous sliding mode controller which follows the above basic concepts. The convergence phase is first sliding mode reaching phase and then sliding phase, the overall controller and the system is fractional-order. In the author’s viewpoint, nonlinearity is included in the proposed controller. The sliding mode controller is nonlinear and we also care about both matched and mismatched disturbances. The buck converter model is also nonlinear.

All the theory mentioned in this paper is from Utkin’s sliding mode control foundation and understood by Filippov sense. That is the key idea of the sliding mode controller and it is developed in this paper. As to the concept ‘co-dimension 2’, the authors do refer to the FV Difonzo’s literature. Unfortunately, it is over mathematical and it seems not related to sliding mode control well. The idea of the paper may guide some multi sliding mode variables sliding mode control of high-order plant, while it may be beyond the author's knowledge and the applications of Buck converter. This paper mainly studies the mathematical model of Buck converter and develops a continuous finite-time sliding mode controller for it. The novelty and the research background of this paper is listed in Section introduction. The overall finite time stability is given strictly and the simulation results show the effectiveness and robustness against unknown upper bounded matched/mismatched disturbances. Thus, the sliding mode method is adequately described and the results of the effectiveness and robustness are clearly shown in Section Simulation. The authors appreciate your suggestion for added literature while as discussed above, the literature cited in this paper does include and follow Utkin’s results. The introduction section mainly states the background of Buck converter controller, the fractional order model of the Buck converter and some different control methods designed for Buck converter. The author thinks the cited references are relevant to the research well. Please check for that.

Reviewer 3 Report

The paper  titled: Continuous adaptive finite-time sliding mode control for fractional-order Buck converter based on Riemann-Liouville definition investigated a continuous adaptive finite-time fractional-order sliding mode  control method for fractional-order Buck converter using the keywordsfractional calculus; Riemann-Liouville; Buck converter; adaptive law; continuous sliding  mode control and  finite-time stability.

1.              What is the main question addressed by the research?

First the authors gave the basic concepts of fractional-order calculus and the relevant lemma.After it the authors dealt with  the  Fractional-order mathematical model of Buck converter based on R-L definition used in this manuscript and discussed the continuous adaptive finite time sliding mode control method, Finally they  carried out some simulations and  presented the concluding remarks supported by the data.

2. Do you consider the topic original or relevant in the field? Does it

address a specific gap in the field?

The topic is relevant  because this paper  proposed  a novel mathematical model based on the Riemann-Liouville fractional calculus definition for the Buck converter that takes into account the existence of non-integer order components and uncertainties/disturbances in practical systems. The simulation results  obtained  demonstrated that the proposed controller can ensure the Buck converter output to track the  reference voltage rapidly and precisely under the influence of matched and mismatched  disturbances.

3. What does it add to the subject area compared with other published

material?

Additionally,  also the adaptive algorithm showed  effectiveness in estimating and handling disturbances. Comparison showed the superiority of the proposed controller.

4. What specific improvements should the authors consider regarding the

methodology? What further controls should be considered?

The authors need to  considered other   controlers to compare them.

5. Are the conclusions consistent with the evidence and arguments presented

and do they address the main question posed?

See question 4.

6. Are the references appropriate? Yes

7. Please include any additional comments on the tables and figures.

The paper is relevant and interesting  and Schematic of controller was done with no errors.

It seems to be original , according to other results from current literature mention on the proposed manuscripit .

 I t deserves  publication after  revisions:

The Numerical simulations  must show the efficiency of the control methods,  as well as the sensitivity of each control strategy to parametric errors.

The authors need to  considered other   controlers to compare them

The sate of the  art  seems be be incomplete   ( see papers published on Mechanical Systems and Signal Processing  journal, MDPI, https://doi.org/10.3390/electronics11010088,  https://doi.org/10.1155/2016/6935081 , and others, and examples)

Author Response

1 The authors need to consider other controllers to compare them.

Thank you very much for your comments here. The author accept your kind advice and add additional simulations to further validate the effectiveness of the proposed controller. Please check new added Figure 7 and 8. The authors design new types of disturbances to test the robustness of the controller. Also, the other methods proposed by some other literature are discussed and the simulation result can be seen clearly that the controller proposed in this paper show superior performance and robustness compared with the others. Please check.

The main revision is shown using red color from page 14 to page 16.

2 The state of the art seems to be incomplete ( see papers published in Mechanical Systems and Signal Processing  journal, MDPI, https://doi.org/10.3390/electronics11010088, https://doi.org/10.1155/2016/6935081, and others, and examples)

Thank you very much for your professional advice here. The authors revise some reference, delete unnecessary literature and add some related literature as the reviewers suggested. Please check for that. The main revision of reference is shown on the last section.

Round 2

Reviewer 1 Report

The revision is satisfied. It can be accepted for publication.

Reviewer 2 Report

Authors replied completely with all the necessary details to the issues raised.